# Model Cautiousness: Towards Safer Deployment in Critical Domains

## Abstract

In this paper, we introduce the concept of model cautiousness, which stresses the importance of aligning a model's confidence with its accuracy in in-distribution (ID) scenarios while adopting a more uncertain approach in out-of-distribution (OoD) contexts. Model cautiousness is framed as a spectrum between justified confidence and complete ignorance, induced by the inability to clearly define a model's domain of expertise. We propose a rigorous post-hoc approach to obtain a cautious model that merges the confidence scores of the primary confidence model and a model discriminating between ID and OoD inputs. A metric to measure the cautiousness error of a confidence model is introduced. We further present a simple method for discriminating ID from OoD inputs and providing a meaningful confidence estimate that an input is OoD. Finally, we benchmark our approach across 12 question-answering and 37 vision datasets, demonstrating its effectiveness in enhancing model cautiousness compared to standard calibration procedures.

## 1 Introduction

As machine learning models become increasingly involved in high-stakes domains, the question of when they are equipped to make critical decisions becomes more pressing. This query lies at the intersection of philosophy and practical application, requiring careful consideration. Intuitively, models that consistently produce accurate predictions in their areas of expertise should exhibit high confidence in their outputs. This is because a strong track record indicates that the model has effectively learned from relevant data and can generalize its knowledge to similar situations.

When a model is asked to make a statement or decision outside its domain of expertise, the situation becomes more complex. The model's opinion may be less valuable in such cases, as it lacks the necessary knowledge and experience to make informed judgments. The question arises: should we trust the model's statement, even if we know it lacks expertise in the given topic? If the model offers a confident opinion outside its expertise and later turns out to be correct, can we justify its initial confidence? Or should it have expressed uncertainty due to its lack of expertise?

Within a model's domain of expertise, it is reasonable to require the confidence exhibited by the model to match the accuracy of the model itself. In other words, the more accurate the model is, the more confident we should expect it to be. It is well established in the literature, however, that even within the model's domain of expertise, models' confidence and accuracy often do not align (Guo et al., 2017). In fact, models often exhibit a largely overconfident behaviour, with a significant risk of incurring into false positive decisions. Among the plethora of uncertainty quantification techniques developed to properly quantify the model's uncertainty (Abdar et al., 2021), calibration stands out as a simple, usually post-hoc, family of techniques which require the model's confidence and accuracy to match (Guo et al., 2017). The confidence of a calibrated model therefore retains a notion of probability, which provides an interpretable measure of risk to the user. However, as we move further from the model's expertise, blindly trusting a model because it exhibits good calibration properties may still be dangerous, as it would encourage to trust models because they happen to be right, without considering whether they have a solid basis upon which to make a statement.

In this paper, we introduce the concept of model cautiousness, which emphasizes the importance of aligning a model's confidence with its accuracy when handling in-distribution (ID) data, while

gradually adopting a more uncertain stance as it encounters out-of-distribution (OoD) data. Cautiousness can be viewed as a spectrum, ranging from justified confidence to complete uncertainty. This gradual transition arises from the inherent inability in defining a model's domain of expertise, making it difficult to distinguish between instances where the model is interpolating within familiar data and those where it is extrapolating beyond its knowledge.

Given the challenges of implementing safety mechanisms in real production environments, particularly when multiple stakeholders control different parts of the product pipeline, we propose an operational definition of cautiousness that avoids imposing overly restrictive requirements for deployment. Similar to calibration, we approach cautiousness as a post-hoc adjustment to the confidence model, allowing it to be developed independently of the primary model. Unlike traditional post-hoc calibration methods, which typically require only confidence scores and target variables from a calibration dataset, our approach additionally leverages the embeddings of calibration inputs to build a model capable of distinguishing between ID and OoD inputs (see Section 3.1). We do not require a specific distribution of OoD data, nor do we rely on the synthetic generation of such data, which could undermine the robustness of the pipeline. However, since qualitative OoD data, even if synthetic, can enhance the ability to differentiate between ID and OoD inputs, we encourage the use of such resources if feasible. Nonetheless, our preferred approach remains agnostic, with minimal assumptions, to lower the barriers to adopting model cautiousness.

In this paper, we make the following contributions:

1. We provide a formal definition of model cautiousness and introduce an error metric to quantify the degree of cautiousness in a confidence model.

2. We propose a simple method for discriminating between ID and OoD inputs, along with a reliable confidence estimate for identifying OoD data.

3. We outline a simple yet rigorous method for achieving model cautiousness by combining the confidence scores from both the primary model and the discrimination model into a cautious confidence score.

4. Finally, we assess the cautiousness of our approach across 37 vision datasets and 12 question-answering datasets. Our results show that our method significantly improves cautiousness compared to models calibrated using standard techniques on the vast majority of datasets.

## 1.1 A Motivating Example

To illustrate the importance of model cautiousness, we begin with a motivating example. Figure 1 (left) illustrates two distinct clusters of data, which we refer to as "moons". The data points in the upper moon are represented in orange, similar to the blob of data located above, while the points in the lower moon are depicted in blue. A classification model is trained using these two moons of data, without any exposure to the blob during the training phase. The model manages to classify perfectly between the two moons. By doing so, it also happens to perfectly classify the out-of-distribution blob of data. The background color indicates the model's confidence, which approaches 1 as we move away from the decision boundary.

Intuitively, the confidence of the classification model is said to be *calibrated* when the confidence scores align with the model's accuracy (for further details, see Section 2). For instance, if the model's accuracy is nearly 1 for certain data, we would anticipate the confidence to also be close to 1. Thus, if we were to assess the calibration error for the blob of OoD data, we would conclude that the model is well calibrated with respect to this distribution, as both confidence and accuracy are nearly 1.

Nevertheless, even though the model is well calibrated, its high confidence for the OoD blob of data may be viewed as undesirable and potentially risky in various practical applications. This concern arises because the model has not been exposed to any data in that specific region of the input space. Ideally, we would prefer a situation akin to Figure 1 (right), where the model demonstrates high confidence near the training data but adopts a more cautious stance as we move further into the OoD area. The left and right images in Figure 1 were generated using histogram binning, a conventional calibration method described in (Zadrozny & Elkan, 2001; Detommaso et al., 2024a), and a *cautiousness* approach, which we will discuss in the following section.

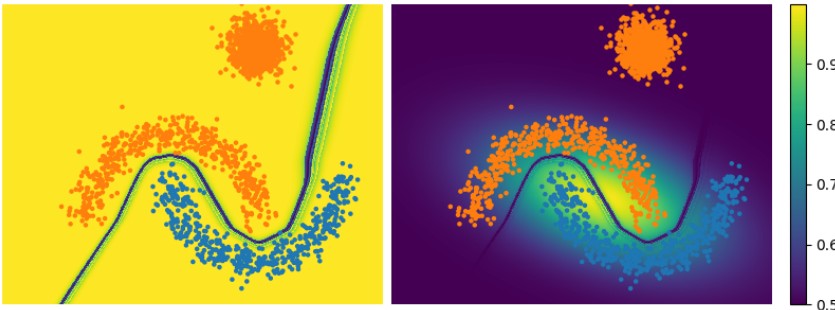

Figure 1: A binary classification model is trained over two moons of data. The background color represents the model's confidence after a calibration (left) and a cautiousness method (right) have been applied, respectively. Althought the model is perfectly calibrated on the blob of OoD data, it also exhibits high confidence OoD, which is undesirable. By contrast, the cautiousness approach requires the model to be uncertain as we move away from the ID data.

## 2 BACKGROUND ON CALIBRATION

Let us denote input variables by $X \in \mathcal{X}$ and target variables by $Y \in \mathcal{Y}$. For simplicity, in this work we restrict the focus to binary targets only, that is $\mathcal{Y} := \{0, 1\}$, but the discussion naturally extends to more general cases. We denote random variables via upper case letters, and reserve lower case letters for their corresponding realizations.

Consider a model $f(x)$ representing the confidence that $Y$ equals 1 for a given input $x$. The following definition introduces calibration, a minimal consistency requirement that endows a confidence model with a notion of probability.

**Definition 2.1** (Calibration). We say that a model $f$ is *calibrated* with respect to the distribution of $(X, Y)$ if and only if

$$\mathbb{P}(Y = 1 | f(X) = p) = p,$$

for all $p \in [0, 1]$ within the support of the distribution of $f(X)$.

In words, calibration requires that the confidence in a positive outcome matches the fraction of times the positive outcome arises, for all confidence levels. If, for instance, we are confidence that an event happens with 80% probability, then the fraction of independent times the event arises, over an infinite number of trials, should be in fact $80\%$. While calibration is often thought of as a distribution-free condition, we remark that Definition 2.1 is equivalent to asking that $Y | f(X) \sim \text{Bernoulli}(f(X))$.

**Definition 2.2** (Expected Calibration Error). We introduce the *Expected Calibration Error* (ECE) defined by

$$\text{ECE}(f) := \mathbb{E}_P\big[\big|\mathbb{P}(Y = 1 | f(X) = P) - P\big|\big], \tag{1}$$

where $P$ is distributed as $f(X)$.

The ECE is a popular measure of calibration error firstly introduced in (Guo et al., 2017). It is immediate to see that if $f$ is calibrated, than the ECE is zero. The ECE has been critized in multiple works for being impossible to compute in practice and not robust to approximations (Błasiok et al., 2023). Several variants have been proposed to improve over the ECE (Nixon et al., 2019). In this work, we will propose a novel definition of calibration error that is built upon the definition of ECE in (1), but we remark that the same concept can be directly applied to any of its variations.

As discussed in Section 1.1, evaluating the ECE against OoD data can lead to scenarios where a model appears perfectly calibrated, resulting in a very low ECE, yet the model may exhibit high confidence in its OoD predictions. This situation can create a misleading sense of security, potentially leading to unsafe deployment of models in critical applications. The next section presents a solution to this issue.

## 3 MODEL CAUTIOUSNESS

In this section, we present the concept of model cautiousness. Essentially, cautiousness requires models to be well-calibrated when operating in-distribution and to exhibit complete uncertainty when dealing with out-of-distribution inputs. We denote the subdomain of out-of-distribution inputs as $\text{OoD} \subset \mathcal{X}$. Importantly, whether an input belongs to OoD or not is a binary random variable because it is impossible to establish a clear boundary between in-distribution and out-of-distribution data. Although we cannot precisely identify where OoD lies, if we were to ascertain that an input is indeed OoD, we would require the model to demonstrate total ignorance regarding the outcome of the target variable. In other words, our predictions in this scenario would reflect the same level of confidence as random guessing. That is,

$$\mathbb{P}(Y = 1 | f(X) = p, X \in \text{OoD}) = \frac{1}{2}, \tag{2}$$

for all $p \in [0, 1]$ in the support of the distribution of $f(X)$. Vice versa, if we do know that the input is not OoD, then a standard calibration requirement should apply, that is

$$\underbrace{\mathbb{P}(Y = 1 | f(X) = p, X \notin \text{OoD})}_{=: \mathcal{F}(p)} = p, \tag{3}$$

again for all $p \in [0, 1]$ in the support of the distribution of $f(X)$.

Let us now introduce a discrimination model $g(x)$ representing the confidence that an input $x$ is OoD. Similarly as for $f$, we ought for the model $g$ to be calibrated, that is the fraction of times an input is OoD should match the discrimination model itself. That is,

$$\underbrace{\mathbb{P}(X \in \text{OoD} | g(X) = q)}_{=: \mathcal{G}(q)} = q, \tag{4}$$

We observe that, by conditioning on whether the input is OoD, we can write

$$\mathbb{P}(Y = 1 | f(X) = p, g(X) = q) = \underbrace{\mathcal{F}(p)(1 - \mathcal{G}(q)) + \frac{1}{2}\mathcal{G}(q)}_{=: \mathcal{H}(p,q)}, \tag{5}$$

which leads us to the following definition.

**Definition 3.1** (Cautiousness). Let us define

$$h(x) := f(x)(1 - g(x)) + \frac{1}{2}g(x). \tag{6}$$

We say that the model $h$ is *cautious* if and only if

$$\mathcal{H}(p, q) = \mathbb{E}[h(X) | f(X) = p, g(X) = q] = p(1 - q) + \frac{1}{2}q, \tag{7}$$

for all $p, q \in [0, 1]$ in the support of the joint distribution of $f(X)$ and $g(X)$.

It is immediate that if $f$ is calibrated ID as in (3) and $g$ is calibrated at discriminating between ID and OoD inputs as in (4), then the model $h$ defined in (6) is cautious. Furthermore, if $f$ is calibrated ID, the pair $(f, g)$ with $g(x) \equiv 0$, also denoted as $(f, 0)$, produces a cautious model $h$ only when evaluated over data that is fully ID, but not otherwise. A standard confidence model $f$ can usually be identified as $(f, 0)$, as it acts under the assumption that all inputs are ID.

We now introduce a measure of cautiousness.

**Definition 3.2** (Expected Cautiousness Error). We introduce the *Expected Cautiousness Error* (ECauE) defined by

$$\text{ECauE}(f, g) := \mathbb{E}_{P,Q}\left[\left|\mathcal{H}(P, Q) - \left(P(1 - Q) + \frac{1}{2}Q\right)\right|\right] \tag{8}$$

Similarly to the ECE, which is an expected absolute error between left- and right-hand sides of the calibration requirement in Definition 2.1, the ECauE is an expected absolute error between left- and right-hand sides of the cautiousness requirements in Definition 3.1. We observe that when all the inputs over which the ECauE is evaluated are ID, then $\text{ECauE}(f, 0)$ reduces to $\text{ECE}(f)$, but the two metrics are different otherwise.

In practice, the ECauE can be estimated using a similar procedure used to estimating the ECE. Given a supervised dataset of inputs $X$, labels $Y$, and binary events for whether $X \in \text{OoD}$, the pairs $(f(X), g(X))$ are jointly binned to a two-dimensional grid. For each joint bin, the values of $\mathcal{F}$ and $\mathcal{G}$ can be empirically estimated, which are in turned used to estimate $\mathcal{H}$. The expression in (8) can then be computed by evaluating the absolute difference and calculate the average weighted by the number of elements in each joint bin.

Algorithm 1 outlines the procedure to obtain a cautious confidence model: given a calibrated model $f$ and a discrimination model $g$, the process involves calculating the model confidence $h$ as defined in equation (6). This confidence can then be utilized to make decisions at an interpretable level of risk, offering a safer alternative to relying solely on calibrated confidence, as it accounts for the model's potential inability to make reliable predictions because outside its area of expertise.

---

**Algorithm 1:** Cautious Confidence Model

---

**Require:** A confidence model $f$, calibration data $D := \{(x_i, y_i)\}_{i=1}^N$.
 1: Use a standard calibration approach to calibrate $f$ with respect to $D$.
 2: Fit a discrimination confidence model $g$ over the calibration data $D$.
 3: Compute the cautious confidence model as in (6).

---

### 3.1 A Discrimination Method

Our approach to construct a model $g$ that is able to discriminate between ID and OoD inputs is pragmatic. Given a calibration dataset, we first fit a multivariate distribution over the embeddings of the calibration inputs. At prediction time, we reconduct the prediction problem to a one-dimensional statistical test, and exploit the p-value to quantify the probability that an input is ID.

Concretely, given the calibration data $D = \{(x_i, y_i)\}_{i=1}^N$, let $z_i$ denote the embeddings of the calibration inputs $x_i$. Because embeddings generally live in an unbounded space, we fit a multivariate Gaussian mixture model, where mean and covariance matrix of the $j$-th Gaussian distribution $N(\cdot | \mu_j, \Sigma_j)$ in the mixture are estimated using all the inputs $x_i$ such that $y_i = j$. As in this work we are restricting the problem to binary classification for simplicity, that is $j = 0, 1$, we have at most two Gaussian distributions in the mixture.

At prediction time, we associate the test input $x$ with the Gaussian in the mixture that is most likely to have drawn the sample, that is $j^* = \arg\max_{j=0,1} N(z | \mu_j, \Sigma_j)$. Under the assumption that $Z \sim N(\cdot | \mu_{j^*}, \Sigma_{j^*})$, we then have $\tilde{Z} := \Sigma^{-\frac{1}{2}}(Z - \mu_{j^*}) \sim N(\cdot | 0, I)$, whence $C := \sum_{k=1}^d \tilde{Z}_k^2 \sim \chi_d^2(\cdot)$, where $d$ denotes the dimension of the input embeddings, and stands for the number of degrees of freedom of the Chi-squared distribution. Then we quantify the probability that an input is OoD by

$$g(x) := \mathbb{P}(C \leq c \,|\, C > q_{1-\alpha}), \tag{9}$$

where $q_{1-\alpha}$ is the $(1-\alpha)$-quantile of a $\chi_d^2$ distribution, and $c$ is a realisation of $C$ obtained starting from the input $x$. The further OoD the sample $x$ is, the larger $c$, the larger the confidence $g(x)$ that the input is OoD. The quantile $q_{1-\alpha}$ works as a guardrail so that whenever $c \leq q_{1-\alpha}$, that is the input is not far enough from the the top of the distribution, we are fully confident that the input is ID, that is $g(x) = 0$.

Algorithms 2 and 3 respectively describes in detail how the fit and confidence prediction functions of the method can be implemented efficiently. We name the method *X2-$\alpha$*, in name of the $\chi_d^2$ distribution used in the algorithm, and the parameter $\alpha$ for the confidence guardrail. Together with a calibrated model $f$, this provides a discrimination model $g$ to eventually compute a cautious model $h$, as described in Algorithm 1.

---

**Algorithm 2:** The X2-$\alpha$ Discrimination Model — Fit

---

**Require:** Calibration data $D := \{(z_i, y_i)\}_{i=1}^{N}$, where $z_i$ denote an embedding of $x_i$; $\alpha \in (0, 1]$.
 1: **for** j=0, 1 **do**
 2:     Fit a Gaussian $N(\cdot|\mu_j, \Sigma_j)$ on the data in $D_j := \{z_i : y_i = j\}$.
 3:     Set $\Sigma_j \mapsto \Sigma_j + 10^{-6} I$ to improve stability.
 4:     Compute the Choleskly factorization $\Sigma_j = L_j L_j^\top$, and the factor inverse $L_j^{-1}$.
 5: **end for**

---

---

**Algorithm 3:** The X2-$\alpha$ Discrimination Model — Predict confidence $g(x)$

---

**Require:** A test embedding $z$.
 1: **for** j=0, 1 **do**
 2:     Compute $\tilde{z}_j := L_j^{-T}(z - \mu_j)$
 3:     Compute $c_j := \tilde{z}_j^\top \tilde{z}_j$
 4:     Evaluate $\log N(z|\mu_j, \Sigma_j) = -\frac{1}{2}(c_j + \log(2\pi)) + \text{sum}\left(\log\left(\text{diag}(L_j^{-1})\right)\right)$
 5: **end for**
 6: Set $j^* := \arg\max_j \log N(z|\mu_j, \Sigma_j)$.
 7: Set $g(x) := \left(1 - \frac{1 - F_{\chi_d^2}(c_{j^*})}{\alpha}\right)^+$, where $F_{\chi_d^2}$ denotes the CDF of a $\chi_d^2$ distribution.

---

## 4 EXPERIMENTS

The purpose of this section is to show the effectiveness of our methods in quantifying model cautiousness, and to show how our method significantly improves cautiousness against just a calibrated model. We benchmark on 37 vision datasets and 13 question-answering datasets - see a detailed list of the datasets in Section 4.1. For every vision dataset, we use a CLIP ViT 32b model (Radford et al., 2021) with Quick GELU (Hendrycks & Gimpel, 2016) to compute confidence scores for each class. For each question-answering dataset, we use a Google Gemma-2b model (Team et al., 2024) to compute confidence scores for each answer. Let us denote by $f_j(X)$ the confidence score associated to the $j$-th class/answer given the image/question $x$, and by $y_j$ a binary variable indicating whether the class/answer $j$ is correct. Because for simplicity in this work we restricted the scope to binary classification, we introduce $\iota(x) := \arg\max_j f_j(x)$, whence we define the confidence score $\tilde{f}(x) := f_{\iota(x)}(x)$ and the corresponding binary target $y := y_{\iota(x)}$.

We form pairs of datasets, of which one is considered as ID and the other OoD. The ID dataset is randomly split 50/50 into calibration and holdout datasets. Each model $\tilde{f}(x)$ is calibrated using the calibration dataset, to produce a final calibrated model $f(x)$. The same calibration dataset is used to fit a discrimination model $g(x)$. We form a test dataset by combining the holdout ID dataset with the OoD in a 50/50 split. Metrics reported in this section are computed over this test dataset unless differently specified. Results are on average over 5 different runs, resulting from 5 different random splits of the ID dataset.

Table 1 compares the ECauE values for $(f, 0)$ and $(f, g)$ across all (ID, OoD) pairs of question-answering datasets, where $g$ represents the confidence of the X2-95 discrimination model. The notation $(f, 0)$ refers to a case where the discrimination model is entirely absent, meaning all inputs are predicted as ID with full confidence. This is equivalent to relying solely on the calibrated model $f$ without any additional information. We observe that in almost every dataset pair, the use of X2-95 results in a notably lower ECauE, demonstrating that the cautious model $h$, formed by combining $f$ and $g$, consistently provides greater cautiousness compared to using the calibrated model $f$ alone.

Similar conclusions hold for pairs of vision datasets, as shown in Figure 2 (left). The figure illustrates that the ECauE decreases significantly when using the X2-95 discrimination model in nearly every dataset pair. Interestingly, although the cautious model prioritizes cautiousness over calibration, it does not consistently worsen the calibration error. In fact, it reduces the ECE in 54% of cases.

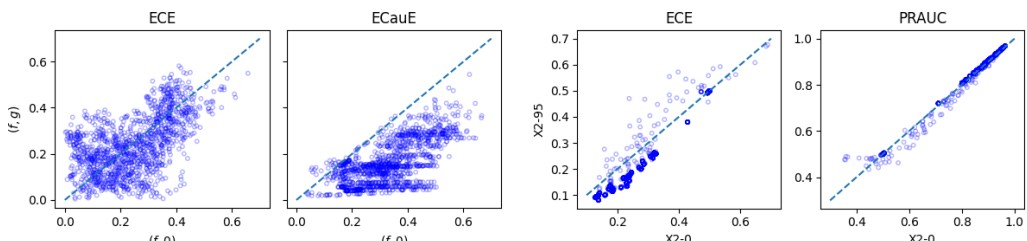

Figure 2: Results in these figures are computed over 37 different vision datasets, on average over 5 runs. The figure on the left shows ECE and ECauE using only a calibrated model $f$ (denoted by $(f, 0)$) versus using a cautious model obtained by the combination of $f$ and X2-95 (denoted by $(f, g)$). The use of $(f, g)$ decreases the ECE around $54\%$ of the times, and it drastically decreases the ECauE. This demonstrates the effectiveness of the cautious model towards increasing cautiousness. The figure on the right compares ECE and PRAUC using X2-0 and X2-95. While results appear fairly similar for the two discrimination models, the X2-95 obtains better ECE and PRAUC than X2-0 respectively around 93% and 83% of the times.

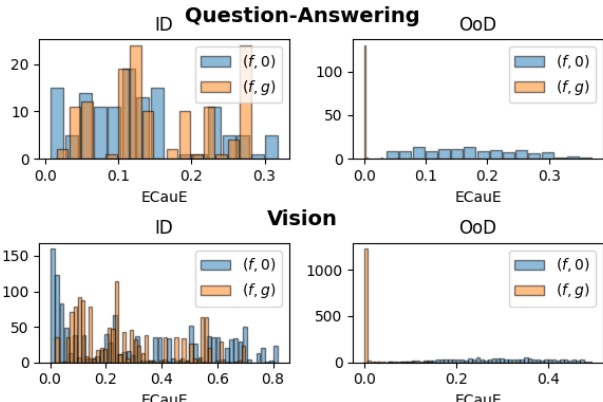

Figure 3: ECauE distribution on question-answering (top) and vision (bottom) test datasets evaluated on fully ID (left) and fully OoD (right) data. Blue vs. orange represent calibrated and cautious models, respectively. We can see that the error in the discrimination model makes the left tail of the ECauE distribution slightly larger for the cautious than for the calibrated model. However, the ECauE for the cautious model is very close to 0 OoD and significantly better than for the calibrated model.

In Figure 2 (right), we also assess the calibration of X2-0 and X2-95. Note that X2-0 is similar to the model proposed in (Venkataramanan et al., 2023), with the key distinction that here we extend the method to use a covariance matrix for each class, rather than a shared one across all data. The results demonstrate that X2-95 consistently provides better calibration (as defined in (4)) compared to X2-0, achieving a lower ECE in 93% of cases and a higher PRAUC in 83% of cases. This indicates that the guardrail quantile introduced in (9) effectively improves the alignment between the discrimination model's confidence and its ability to distinguish between ID and OoD inputs.

In Figure 3, we present the ECauE distribution for question-answering (top) and vision (bottom) test datasets, evaluated on fully ID (left) and fully OoD (right) data. We compare the errors of $(f, g)$ and $(f, 0)$, where X2-95 is used as the discrimination model $g$. It is worth noting that when all the data are ID, the optimal choice for the discrimination model is $g(x) \equiv 0$, meaning any generic model $g$ can only perform worse. However, we observe that the error introduced by $(f, g)$ compared to $(f, 0)$ mostly shifts the left tail of the ECauE distribution slightly higher, without significantly affecting the overall distribution. Conversely, when the test data is fully OoD, $g$ significantly reduces the error, highlighting the benefit of using a cautious model over a calibrated one when we do not know whether the data is ID or OoD.

| ID dataset | OoD dataset | ECauE$(f,0)$ | ECauE$(f,g)$ |
|---|---|---|---|
| anli | xcopa | 0.194873 | **0.086460** |
| | bigbench-mc | 0.218318 | **0.128833** |
| | xnli | 0.191277 | **0.109544** |
| | piqa | 0.212760 | **0.101698** |
| | prost | 0.161699 | **0.128043** |
| | True_mathqa | 0.160231 | **0.108828** |
| | truthfulqa_mc1 | 0.178957 | **0.118723** |
| | winogrande | 0.191063 | **0.141359** |
| | openbookqa | 0.175881 | **0.116813** |
| | mc_taco | 0.198397 | **0.146840** |
| | race | 0.208803 | **0.123891** |
| xcopa | anli | 0.051657 | **0.026142** |
| | bigbench-mc | 0.053268 | **0.024751** |
| | xnli | 0.088327 | **0.057906** |
| | piqa | 0.113961 | **0.047305** |
| | prost | 0.071667 | **0.036261** |
| | True_mathqa | 0.067857 | **0.044478** |
| | truthfulqa_mc1 | 0.098448 | **0.041711** |
| | winogrande | 0.096988 | **0.043057** |
| | openbookqa | 0.102907 | **0.045866** |
| | mc_taco | 0.103338 | **0.044193** |
| | race | 0.113378 | **0.047256** |
| bigbench-mc | anli | 0.176408 | **0.064507** |
| | xcopa | 0.277806 | **0.109804** |
| | xnli | 0.255108 | **0.124373** |
| | piqa | 0.320390 | **0.143959** |
| | prost | 0.226711 | **0.149327** |
| | True_mathqa | 0.226108 | **0.151922** |
| | truthfulqa_mc1 | 0.295069 | **0.153045** |
| | winogrande | 0.280228 | **0.154819** |
| | openbookqa | 0.295305 | **0.159965** |
| | mc_taco | 0.280716 | **0.159126** |
| | race | 0.310790 | **0.159105** |
| xnli | anli | 0.152319 | **0.057046** |
| | xcopa | 0.176577 | **0.153925** |
| | bigbench-mc | 0.167059 | **0.060496** |
| | piqa | 0.174088 | **0.063226** |
| | prost | 0.152066 | **0.062844** |
| | True_mathqa | 0.158880 | **0.062579** |
| | truthfulqa_mc1 | 0.157442 | **0.065173** |
| | winogrande | 0.149253 | **0.069914** |
| | openbookqa | 0.151942 | **0.062421** |
| | mc_taco | 0.151171 | **0.074925** |
| | race | 0.159429 | **0.070791** |
| piqa | anli | **0.091851** | 0.118873 |
| | xcopa | 0.302141 | **0.174622** |
| | bigbench-mc | 0.145058 | **0.139208** |
| | xnli | 0.161596 | **0.145675** |
| | prost | **0.138841** | 0.143589 |
| | True_mathqa | **0.152763** | 0.155030 |
| | truthfulqa_mc1 | 0.230172 | **0.163187** |
| | winogrande | 0.199415 | **0.157626** |
| | openbookqa | 0.232821 | **0.167207** |
| | mc_taco | 0.160972 | **0.156282** |
| | race | 0.257574 | **0.168609** |
| prost | anli | 0.178440 | **0.044172** |
| | xcopa | 0.226167 | **0.052584** |
| | bigbench-mc | 0.182836 | **0.053108** |
| | xnli | 0.162214 | **0.053458** |
| | piqa | 0.226207 | **0.057549** |
| | True_mathqa | 0.147478 | **0.055795** |
| | truthfulqa_mc1 | 0.189060 | **0.060452** |
| | winogrande | 0.165478 | **0.059089** |
| | openbookqa | 0.183577 | **0.064157** |
| | mc_taco | 0.149152 | **0.053459** |
| | race | 0.195267 | **0.057960** |

| ID dataset | OoD dataset | ECauE$(f,0)$ | ECauE$(f,g)$ |
|---|---|---|---|
| True_mathqa | anli | 0.166498 | **0.091963** |
| | xcopa | 0.191961 | **0.121566** |
| | bigbench-mc | 0.138799 | **0.112402** |
| | xnli | 0.138327 | **0.117819** |
| | piqa | 0.179438 | **0.125168** |
| | prost | **0.121034** | 0.121657 |
| | truthfulqa_mc1 | 0.148316 | **0.125165** |
| | winogrande | 0.128156 | **0.123862** |
| | openbookqa | 0.141484 | **0.126466** |
| | mc_taco | **0.116892** | 0.124014 |
| | race | 0.152895 | **0.125060** |
| truthfulqa_mc1 | anli | 0.226312 | **0.147654** |
| | xcopa | **0.166179** | 0.179006 |
| | bigbench-mc | 0.202163 | **0.162108** |
| | xnli | 0.195775 | **0.167246** |
| | piqa | **0.166895** | 0.187132 |
| | prost | 0.229024 | **0.154224** |
| | True_mathqa | 0.239757 | **0.156519** |
| | winogrande | 0.198708 | **0.164725** |
| | openbookqa | 0.184377 | **0.172840** |
| | mc_taco | 0.191781 | **0.167848** |
| | race | **0.175773** | 0.176841 |
| winogrande | anli | **0.055605** | 0.065443 |
| | xcopa | 0.285250 | **0.145813** |
| | bigbench-mc | **0.092341** | 0.103105 |
| | xnli | 0.186673 | **0.126741** |
| | piqa | 0.307480 | **0.150027** |
| | prost | 0.145653 | **0.111462** |
| | True_mathqa | 0.145136 | **0.120347** |
| | truthfulqa_mc1 | 0.256190 | **0.140062** |
| | openbookqa | 0.247371 | **0.139319** |
| | mc_taco | 0.211634 | **0.144226** |
| | race | 0.269374 | **0.141792** |
| openbookqa | anli | 0.183179 | **0.139512** |
| | xcopa | 0.194433 | **0.149510** |
| | bigbench-mc | 0.208411 | **0.128750** |
| | xnli | 0.197463 | **0.141157** |
| | piqa | 0.201149 | **0.158886** |
| | prost | 0.209307 | **0.140694** |
| | True_mathqa | 0.210884 | **0.139551** |
| | truthfulqa_mc1 | 0.193029 | **0.149191** |
| | winogrande | 0.194993 | **0.143232** |
| | mc_taco | 0.204869 | **0.136131** |
| | race | 0.192379 | **0.148286** |
| mc_taco | anli | 0.048849 | **0.025789** |
| | xcopa | 0.229644 | **0.026954** |
| | bigbench-mc | 0.165212 | **0.026435** |
| | xnli | 0.106443 | **0.025512** |
| | piqa | 0.245675 | **0.030207** |
| | prost | 0.060963 | **0.024790** |
| | True_mathqa | 0.079396 | **0.026713** |
| | truthfulqa_mc1 | 0.155009 | **0.031477** |
| | winogrande | 0.101956 | **0.029712** |
| | openbookqa | 0.144715 | **0.033470** |
| | race | 0.172014 | **0.031330** |
| race | anli | 0.267539 | **0.074490** |
| | xcopa | 0.241411 | **0.148867** |
| | bigbench-mc | 0.215293 | **0.112469** |
| | xnli | 0.213885 | **0.125642** |
| | piqa | 0.247089 | **0.147459** |
| | prost | 0.241692 | **0.100604** |
| | True_mathqa | 0.257653 | **0.089054** |
| | truthfulqa_mc1 | 0.232243 | **0.129010** |
| | winogrande | 0.214164 | **0.124248** |
| | openbookqa | 0.226671 | **0.128183** |
| | mc_taco | 0.210200 | **0.118511** |

Table 1: Results in these tables are computed over 12 different text datasets on average over 5 runs. Each row in the tables is computed over a dataset obtained by a 50/50 split between the ID and the OoD dataset. The tables show the ECauE using only a calibrated model $f$ (denoted by $(f,0)$) versus using a cautious model obtained by the combination of $f$ and X2-95 (denoted by $(f,g)$). The use of $(f,g)$ decreases the ECauE in almost all rows, demonstrating the significantly better cautiousness of the cautious model obtained from $(f,g)$ compared to just the calibrated model $f$.

### 4.1 DATASETS

We list here the datasets used for the experiments of Section 4.

Question-answering: ANLI (Nie et al., 2019), XCOPA (Ponti et al., 2020), BIG-Bench (bench authors, 2023), XNLI (Conneau et al., 2018), PIQA (Bisk et al., 2020), PROST (Aroca-Ouellette et al., 2021), (Amini et al., 2019), TruthfulQA (Lin et al., 2021), Winogrande (Sakaguchi et al., 2021), OpenbookQA (Mihaylov et al., 2018), MC-TACO (Zhou et al., 2019), RACE (Lai et al., 2017).

Vision: MNIST (LeCun et al., 1998), FER2013 (Goodfellow et al., 2013), SVHN (Netzer et al., 2011), PCAM (Veeling et al., 2018), FGVC-Aircraft (Maji et al., 2013), KITTI (Geiger et al., 2012), ImageNet-O (Hendrycks et al., 2021b), ImageNet-A (Hendrycks et al., 2021b), ImageNet-R (Hendrycks et al., 2021a), ImageNetV2 (Recht et al., 2019), ImageNet-Sketch, ImageNet1k (Russakovsky et al., 2015), (Wang et al., 2019), EuroSAT (Helber et al., 2019), STL-10 (Coates et al., 2011), Caltech-101 (Fei-Fei et al., 2004), Oxford-IIIT Pet (Parkhi et al., 2012), Dmlab (Zhai et al., 2019), smallNORB (LeCun et al., 2004), CIFAR10, (Krizhevsky et al., 2009), CIFAR100 (Krizhevsky et al., 2009), Oxford 102 Flower (Nilsback & Zisserman, 2008), SUN397 (Xiao et al., 2010), Rendered SST2 (Socher et al., 2013), CLEVR (Johnson et al., 2017), ObjectNet (Barbu et al., 2019), PASCAL VOC (Everingham et al.), Diabetic Retinopathy Kaggle & EyePacs (2015), dSprites (Higgins et al., 2017), DTD (Cimpoi et al., 2014), Stanford Cars (Krause et al., 2013), Country211 (Radford et al., 2021), GTSRB (Stallkamp et al., 2012), RESISC45 (Cheng et al., 2017).

## 5 RELATED WORK

Several works in the literature focus on providing calibration or conformal prediction guarantees under covariate shift assumptions (Tibshirani et al., 2019; Park et al., 2020; Jonkers et al., 2024). This is relevant to our work since shifted covariates are considered OoD. However, these approaches differ fundamentally as they aim to ensure OoD calibration, while the goal of cautiousness is to ensure that the system remains uncertain when faced with OoD inputs. Additionally, the success of these methods often depends on estimating a density ratio, which can be unreliable in practice (Sugiyama et al., 2010).

Another branch of the literature focuses on OoD detection. In ODIN (Liang et al., 2017; Hsu et al., 2020), the authors attempt to detect OoD inputs using a softmax approach with temperature scaling (Guo et al., 2017) and input preprocessing. Other works use distance functions to measure how far an input is from the training distribution, with the Mahalanobis distance (Lee et al., 2018) being the most common. This distance is equivalent to the negative log-probability density function (PDF) of a Gaussian distribution. A related approach, the relative Mahalanobis distance (Ren et al., 2021), computes the maximum ratio between the Gaussian likelihood for a specific class and the likelihood for the entire dataset. Other works based on the nearest neighbor distance (Sun et al., 2022; Detommaso et al., 2022) calculate the distance between an input and the closest training input, but these methods require knowledge of the training inputs and can be computationally expensive. DUQ (Van Amersfoort et al., 2020) introduces a radial basis function to estimate confidence, while DDU (Mukhoti et al., 2023) uses a Gaussian Mixture Model (GMM) over classes, similar to our approach in Section 3.1. In (Venkataramanan et al., 2023), a GMM with a shared covariance matrix is used, and OoD confidence is derived using a Chi-squared approach. This method is similar to X2-$\alpha$, but we utilize separate covariance matrices as in (Mukhoti et al., 2023), and introduce a guardrail quantile in (9), which is shown in Section 4 to improve performance. Another related line of work is SNGP (Liu et al., 2020), where the authors introduce a decomposition similar to ours in (5). However, they do not impose a full uncertainty requirement for OoD inputs as we do in (2), and their approach is not post-hoc, requiring modifications to the confidence model.

## 6 CONCLUSION

In this work, we introduced the concept of model cautiousness, which requires a model to be calibrated for in-distribution (ID) data while remaining uncertain for out-of-distribution (OoD) inputs. The balance between these two states is rigorously determined based on the performance of an auxiliary model that discriminates between ID and OoD inputs. We also introduced a metric to measure cautiousness error and proposed a method for discriminating ID versus OoD inputs, which pro-

vides meaningful confidence estimates. Our approach was evaluated across numerous vision and question-answering datasets, showing its effectiveness in improving model cautiousness compared to standard calibration methods.

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
