# OpenReview forum: "Model Cautiousness: Towards Safer Deployment in Critical Domains"
_ICLR.cc/2025/Conference — ICLR 2025 Conference Withdrawn Submission_

### Official Review · Reviewer_9e7R · 2024-10-16

**Soundness:** 3
**Presentation:** 3
**Contribution:** 1
**Rating:** 3
**Confidence:** 2

**Summary:**

The paper presents several (somewhat independent) technical contributions:
(C1) A new concept, "cautiousness", that refers to a model being calibrated on ID data while being uncertain on OOD data.
(C2) A method to recognize ID vs OOD inputs.
(C3) A method to achieve their concept of "cautiousness" by combining the uncertainty of a given model with the method (2).

**Strengths:**

- Clear writing.

- Extensive evaluation, very large number of datasets and models.

**Weaknesses:**

My concerns/questions are at a high level, regarding the motivation for this work and the differences with existing concepts. I may be missing something, so I'm keen to update my score if the authors can clarify the points below. In any case, the paper will need clarifications about these points.

- (W1) The related work says that this work differs fundamentally from existing work that aims to ensure OOD calibration, because this one is about being always uncertain when OOD (if I understand correctly). But the former seems to be much more useful, and make more sense for the following reason. "OOD" is not a binary property as correctly noted in Section 3). For example, in the case of covariate shift, one cannot strictly classify one example as ID or OOD, and with images, since they're high dimensional, any unseen test example is outside the convex hull of training examples. This is why it seems to make more sense to aim for the same thing (calibration) whether ID and OOD, rather than aiming for complete uncertainty whenever OOD (as proposed, if I understood correctly?).

- (W2) The contribution C2 seem to have a lot in common with existing work on OOD detection, which is a large, well-established area of research. The related work mentions several existing methods, but the field is much larger than that. It discusses some technical similarities and differences with a few existing methods, but it does **not** discuss the need for a new method (C2) . The proposed method (C2) is presented as "a model that discriminates between ID and OoD inputs". But to me this is exactly what OOD detection is. So why don't the authors just call "a new OOD method"? I may be missing something. But in any case, the need for this method (given the plethora of existing ones) should be made clearer.

- (W3) The need for the contribution C1 is not clear to me. It feels like putting a new name (and defining a combined metric) for two existing areas (calibration and OOD detection). Is the concept of "cautiousness" an academic curiosity, or does it answer a need of ML practitioners in the evaluation and/or deployment of ML models? Since this paper seems to address a very practical problem, I think the authors should first describe how the challenges of calibration/OOD inputs are handled in current industrial ML deployments, and then discuss which issues in these practices need new solutions. The authors do provide a "motivating example" in the introduction, but the two-moon dataset did not convince me at all that the paper was adressing a concrete necessity.

**Questions:**

Please clarify each of the 3 weaknesses above. Thanks!

The PDF does not have hyperlinks on sections/equation references nor on citations. This makes the reading **extremely** tedious.

---

### Official Review · Reviewer_3V27 · 2024-11-03

**Soundness:** 1
**Presentation:** 2
**Contribution:** 1
**Rating:** 3
**Confidence:** 4

**Summary:**

This paper introduces a new concept of “cautiousness” which measures two things: how calibrated a model is on ID and how close to random guess it is on OOD. A measure of cautiousness - ECauE is proposed as well as an algorithm to compute it. In addition to the base classifier f this algorithm requires training a separate OOD detection model g and ECauE is computed jointly for f and g.

**Strengths:**

I could not find any strengths in this paper, unfortunately.

**Weaknesses:**

## Questionable motivation
This paper is trying to solve a task of making a model calibrated on ID while making random predictions on OOD at the same time.
I do not understand why this task needs to be solved at the first place. It is way too complex because it entangles two unsolved problems: model calibration and OOD detection. I believe it is better to solve these problems separately.

## Unrealistic assumptions
To solve the proposed problem authors make very unrealistic assumptions:
- For simplicity, they assume that the classification task is binary (line 128), which is not the case for the most of the real world scenarios (e.g. ImageNet for computer vision). Could the authors please explain whether it is possible to remove this assumption and keep the theory behind the method work (I am asking because results are computed for ImageNet and other non-binary datasets in Table 1 by redefining Y as an indicator of correct prediction)?
- They assume that embeddings of ID data follow normal distribution (lines 255-256) while embeddings of OOD data don't. Could the authors please provide any evidence that justifies this for realistic cases (e.g. ImageNet)?

## Flaws in crucial definitions
- The definition for target labels in classification is not properly given. Namely, it is not clear whether "y" is class label or probability that model is correct (as said in line 303).
- It is not well defined what kind of data is considered ID and OOD in this paper.
- It is not clear how models are calibrated on ID. It is a crucial part of the ECauE computation, but I only found the phrase: “use a standard calibration approach” in line 236. Same goes for details of fitting g(x) model.
- H(p, q) is divergently defined as probability in Eq. 5 line 192 and as an expectation in line 202.
- Why is expectation needed in Eq. 7, isn't the following always true: h(x) = p(1-q) + 1/2q when f(x) = p, g(x) = q? It is confusing, because it seems that this equation always holds, and therefore, all models can be called "cautious" according to the given definition.
- I don't understand how eq. 5 is made, let's define A := (Y=1), B := x is OOD, C := (f(x) = p), D := (g(x) = q), then after conditioning on B we have: P(A | C, D) =  P(A | C, D, not B)*(1 - P(B)) + P(A | C, D, B)*P(B), while you say that P(A | C, D) = P(A | C, not B) * (1 - P(B | D)) + P(A | C, B) * P(B | D).

## Weak empirical results
- The main results (Table 1 for QA and Figure 2 for vision) don't provide the downstream task performance (QA results also lack ECE). Performance should be provided because it is important to remember that we care about calibration and random predictions on OOD only when models capable of solving the downstream task, ECE should be provided to verify that lower ECauE does not break the model calibration.
- The main results for vision provide ECE and show that adding OOD detection model can increase as well as sometimes decrease ECE (see Figure 2 left - one point might have higher ECE for (f,g) than for (f,0) as well as lower). Such an increase of ECE violates the goal of the proposed method: keep models calibrated on ID.

## Computational limitations
I think that the proposed method is not usable in practice due to its computational burden. The suggested method for computing ECauE - a measure of "cautiousness" - requires training an OOD detection model g in addition to the base model f (see Algorithm 1). To compute ECE, for example, no additional models need to be trained.

## Lack of explanation
- Why do you need Cholesky decomposition in Algorithm 2?

## Unclear writing
PRAUC in line 338 is not defined.

## Typos

- 142: we are confidence
- 223: in turn
- 266: describes
- 276: Choleskly → Cholesky

**Questions:**

- Why is Bernoulli formulation in line 145 needed?
- In Figure 3 EcauE is low on OOD and high on ID, should not it be vice versa, i.e. models are well calibrated on ID and not calibrated on OOD.

**Details Of Ethics Concerns:**

I don't have any ethics concerns

---

### Official Review · Reviewer_Q5Ze · 2024-11-04

**Soundness:** 2
**Presentation:** 2
**Contribution:** 2
**Rating:** 3
**Confidence:** 4

**Summary:**

- Motivating Claim: It is desirable that models be calibrated on in-distribution data and exhibit higher (perhaps total) uncertainty on OOD data.
- A notion called "cautiousness" is introduced, that is a combination of ECE on IID data and requires high uncertainty on OOD data.
- The authors define the ECauE (expected cautiousness error) metric.
- The authors introduce a method for fitting an cautiousness estimator atop a pretrained model.
- They show that the estimator they define (X2-95) achieves a better score on the metric (ECauE) they define.

**Strengths:**

- The concept that uncertainty over OOD data should be treated differently than uncertainty over IID data is a neat, systems level way of thinking about model safety.
- The experimental evaluation has broad coverage of datasets for 2 tasks and 1 model for each task.

**Weaknesses:**

### Metric is not justified strongly and makes the paper's story circular
The paper uses circular reasoning in the sense that the authors define a metric, and then propose a method which improves over a baseline w.r.t to the proposed metric. The only attempt made to justify their formulation of the metric is briefly, informally, in the introduction. The argument given is roughly that "we think this is how models should behave, and here is a metric that captures whether they behave that way". **What is the motivation for the metric other than the opinion of the authors? Is there some concrete benefit to models behaving this way?**


### No use of contemporary OOD / Calibration work
The metric basically comes down to being calibrated for IID inputs and being uncertain for OOD inputs. There is a well-established line of research on OOD detection. There is also a well established line of research on model calibration. This suggests a natural baseline, which is to stick well-known OOD detection baseline in front of a model that some well known calibration method has been applied to and simply hardcode low confidence for OOD inputs and calibrated confidence for IID inputs. Why not do this? I don't get the point of introducing some new method when tons of OOD detection methods already exist. There are also no experiments done to show how the new OOD detection method does compared to existing OOD detection works.

### Operational use of OOD seems unjustified
Take Table 1. What makes the datasets you've chosen to be OOD actually OOD for the model? Shouldn't OOD be w.r.t to the pretraining or at least instruction tuning distribution of the model? I don't get how you can say that xyz dataset is OOD for a model. **I don't think calibrating a LLM on one dataset makes a bunch of other datasets OOD for that LLM — what if the LLM was trained on similar data?**

### Selective prediction already answers the motivation, but no mention in paper
Selective prediction is a well-established and active line of work (https://arxiv.org/abs/1901.09192, https://arxiv.org/abs/2306.08751, https://research.google/blog/introducing-aspire-for-selective-prediction-in-llms/). Selective prediction requires a model to abstain when it does not know the answer to a query. A model with perfect risk vs coverage for selective prediction would never make mistakes — if it answered, whether on OOD or IID settings, it would be right. It's a strictly stronger notion than ECE, IMO. What does the introduced metric offer that selective prediction does not? Ultimately I don't think we care about metrics for the sake of metrics. We care about what they can be used for. If this metric could be used for selective prediction, then I could see a nice use case for it. Otherwise, it just seems to be a roundabout way of doing OOD detection. **If a large, pretrained model has really good accuracy and calibration on OOD data, is the data really OOD?**

### Summary
The problem with the work is that it takes a well-known notion (ECE) and says that a model having good ECE on OOD data is bad (why...?) and then develop a _new_ method for OOD detection without comparing it to others (why?). IMO having good ECE on OOD data suggests that the data is not really OOD. In any case, you can't define OOD for a large pretrained model declaratively by saying "this data is OOD", there _must be a reason that model is OOD other than you saying it is_ — especially since these models have probably been trained on similar data to standard benchmarks (e.g. https://arxiv.org/abs/2406.04244v1). Finally, **I think selective prediction already subsumes the use case for this metric without needing to reference the notion of OOD data** (which I think is the big weakness in the argument of the paper): we simply want the model to know when it knows the answer and when it does not know.

**Questions:**

Please see the weaknesses section, I have listed a detailed write up of my issues with the paper there.

---

### Official Review · Reviewer_ojM4 · 2024-11-05

**Soundness:** 3
**Presentation:** 3
**Contribution:** 2
**Rating:** 6
**Confidence:** 4

**Summary:**

This paper introduces the concept of model cautiousness, encouraging the model to maintain high uncertainty when inferring out-of-distribution (OOD) samples, even if its inference results are correct. To achieve this, the proposed cautious confidence model decouples the confidence generation model from the discrimination model, and formulates the detection of an OOD sample as a binary event. Additionally, a new metric called Expected Cautiousness Error (ECauE) is proposed to evaluate the model's performance in terms of both caution and calibration. The proposed method is evaluated on 12 question-answering and 37 vision datasets. Based on the results, the paper claims that the method demonstrates superior model cautiousness compared to standard calibration procedures.

**Strengths:**

- Paper Writing: The paper has a coherent narrative and logical flow. The motivation for introducing the concept of model cautiousness is well-explained, with examples that illustrate the concept effectively. The schematic illustrations are clear and well-designed, allowing readers to quickly grasp the main ideas.
- Innovative Topic: The study of model cautiousness is both intriguing and innovative, offering a new perspective on evaluating a model’s confidence in its inferred outcomes.
- Reasonable Proposal: The proposed cautious confidence model and its component design align well with the formulated problem and are supported by theoretical foundations.
- Evaluation: The proposed method is evaluated on an extensive range of datasets, consistently demonstrating superior results compared to standard calibration procedures.

**Weaknesses:**

- Problem Setting: While I acknowledge the innovation of this paper's topic, I feel that formulating OOD detection as a binary event is overly simplistic. Intuitively, it seems necessary to consider different OOD scenarios, such as samples that are not part of the training distribution but belong to the same category (as in the evaluations in this paper) versus samples from entirely different categories. Additionally, OOD literature distinguishes between near-OOD and far-OOD settings. If the problem formulation considered multiple OOD types rather than treating it as a binary event, it could potentially be more impactful and offer deeper insights.
- Lacking Baselines: Although the proposed method is compared to standard calibration procedures across extensive datasets and shows superior performance, it lacks a comparison with baselines that also emphasize model cautiousness. Such a comparison would better demonstrate the proposed method's effectiveness in advancing model cautiousness.
- Missing Definitions: Some notations or abbreviations are undefined, such as PRAUC. I assume this refers to AUPRC (area under the precision-recall curve), which is more commonly used.
- No Discussion on Limitations: The paper lacks a paragraph that comprehensively discusses potential limitations, either in the problem formulation or in the proposed method.

**Questions:**

- Can the formulated problem easily extend to handle a broader range of OOD situations?
- Are there any baselines that also account for model cautiousness? If so, does the proposed method still achieve superior performance compared to those baselines?
- What are the limitations of the proposed method or the formulated problem?

---

### Note · Authors · 2024-11-12

**Comment:**

Given the ratings, I withdraw this paper.

**Withdrawal Confirmation:**

I have read and agree with the venue's withdrawal policy on behalf of myself and my co-authors.